# The Role of Treatment-Related Parameters and Brain Morphology in the Lesion Volume of Magnetic-Resonance-Guided Focused Ultrasound Thalamotomy in Patients with Tremor-Dominant Neurological Conditions

**DOI:** 10.3390/bioengineering11040373

**Published:** 2024-04-12

**Authors:** Rosa Morabito, Simona Cammaroto, Annalisa Militi, Chiara Smorto, Carmelo Anfuso, Angelo Lavano, Francesco Tomasello, Giuseppe Di Lorenzo, Amelia Brigandì, Chiara Sorbera, Lilla Bonanno, Augusto Ielo, Martina Vatrano, Silvia Marino, Alberto Cacciola, Antonio Cerasa, Angelo Quartarone

**Affiliations:** 1IRCCS Centro Neurolesi “Bonino Pulejo”, 98124 Messina, Italy; rosa.morabito@irccsme.it (R.M.); simona.cammaroto@irccsme.it (S.C.); militiannalisa@gmail.com (A.M.); chiara.smorto@irccsme.it (C.S.); carmelo.anfuso@irccsme.it (C.A.); amelia.brigandi@irccsme.it (A.B.); chiara.sorbera@irccsme.it (C.S.); lilla.bonanno@irccsme.it (L.B.); augusto.ielo@irccsme.it (A.I.); silvia.marino@irccsme.it (S.M.); 2Mater Domini University Hospital, Magna Graecia University, 88100 Catanzaro, Italy; lavano@unicz.it; 3Humanitas Clinical Institute of Catania, 95045 Catania, Italy; ftomasel@unime.it; 4S. Anna Institute, 88900 Crotone, Italy; martinavatrano92@gmail.com; 5Brain Mapping Lab, Department of Biomedical, Dental Sciences and Morphological and Functional Images, University of Messina, 98122 Messina, Italy; alberto.cacciola0@gmail.com; 6Institute for Biomedical Research and Innovation (IRIB), National Research Council of Italy (CNR), 98164 Messina, Italy; 7Pharmacotechnology Documentation and Transfer Unit, Preclinical and Translational Pharmacology, Department of Pharmacy, Health Science and Nutrition, University of Calabria, 87036 Arcavacata, Italy

**Keywords:** MRgFUS, tremor, thalamotomy, n° of transducer elements, maximal mean temperature reached

## Abstract

Purpose: To determine the best predictor of lesion volume induced by magnetic resonance (MR)-guided focused ultrasound (MRgFUS) thalamotomy in patients with tremor-dominant symptoms in Parkinson’s disease (PD) and essential tremor (ET) patients. Methods: Thirty-six neurological patients with medication-refractory tremor (n°19 PD; n°17 ET) were treated using a commercial MRgFUS brain system (Exablate Neuro 4000, Insightec) integrated with a 1.5 T MRI unit (Sigma HDxt; GE Medical System). Linear regression analysis was used to determine how the demographic, clinical, radiological (Fazekas scale), volumetric (total GM/WM/CSF volume, cortical thickness), and MRgFUS-related parameters [Skull Density Ratio (SDR), n° of transducer elements, n° of sonications, skull area, maximal energy delivered (watt), maximal power delivered (joule), maximal sonication time delivered, maximal mean temperature reached (T°C_max), accumulated thermal dose (ATD)] impact on ventral intermediate (VIM)-thalamotomy-related 3D volumetric lesions of necrosis and edema. Results: The VIM thalamotomy was clinically efficacious in improving the tremor symptoms of all the patients as measured at 1 week after treatment. Multiple regression analysis revealed that T°C_max and n° of transducer elements were the best predictors of the necrosis and edema volumes. Moreover, total WM volume also predicted the size of necrosis. Conclusions: Our study provides new insights into the clinical MRgFUS procedures that can be used to forecast brain lesion size and improve treatment outcomes.

## 1. Introduction

Neurological tremors are a common symptom of several neurological disorders, including essential tremor (ET), Parkinson’s disease (PD), and multiple sclerosis. These tremors can profoundly impact patients’ quality of life and hinder their ability to perform daily activities. While conventional treatments such as medications, deep brain stimulation, radiofrequency, and radiosurgery thalamotomy are available, they may not yield the desired results for all patients and can entail significant side effects [1].

A promising advancement in the field is magnetic-resonance-guided focused ultrasound (MRgFUS) thalamotomy. This innovative, non-invasive technique targets specific structures using high-intensity focused ultrasound beams, generating thermal ablative necrosis. In the setting of neurological tremors, the thermal ablation of the ventral intermediate (VIM) thalamic nucleus disrupts the transmission of tremor signals from cerebellar input to the motor cortex, thus inducing an immediate reduction in clinical symptoms. Furthermore, sustained clinical benefits have been shown to persist both at 3 and 12 months from intervention, with a mean improvement in the hand tremor score of 47% and 40%, respectively [2].

Intraoperative MRI procedures and the related parameters are fundamental for performing precise anatomical targeting, as well as for planning and monitoring the induced number of sonications required to achieve therapeutic ablation [3]. Indeed, in order to achieve long-lasting clinical effects, an appropriately sized lesion must be achieved [4]. The different amounts of supplied acoustic energy required to produce a lesion in the brain via MRgFUS vary across patients, depending on the acoustic properties and the density of every single intervening tissue (white matter, gray matter, and cerebral spinal fluid), which influences the amount of energy carried by the acoustic wave that is transmitted rather than reflected or absorbed [5,6,7]. The skull bone is the main barrier for ultrasounds traversing the brain, and its inter-subject variability in thickness and density represents one of the main features leading to the heterogeneous magnitude of the applied acoustic power and treatment time. To evaluate the skull’s ultrasound permeability, the skull density ratio (SDR) is typically assessed prior to treatment [8]. High SDR values (SDR > 0.45) showed a positive relationship with the maximal mean temperature reached (T°C_max), whereas low SDR values (SDR < 0.40) may pose technical challenges, such as a prolonged treatment duration or increased energy requirements [9]. Recent findings demonstrated that MRgFUS treatment for ET can be effectively and safely performed in patients with an SDR < 0.40, although it yields superior outcomes when the SDR ≥ 0.45 [10]. In contrast, Sinai et al. [11] observed no statistically significant difference in the mean SDR between patients experiencing sustained symptom improvement and those facing symptom recurrence in a trial including patients with an SDR < 0.40.

Another fundamental parameter for a successful procedure is the rate of temperature change between the neuromodulation step and thermal ablation. Generally, at least two sonications at a temperature of 56 °C are required to achieve precise necrosis [12]. Hence, defining the causes of the temperature increase is essential to enhance the effectiveness of such treatments [13]. With regard to lesion size prediction in repeated sonications, the accumulated thermal dose (ATD) has been demonstrated to predict the lesion sizes in ET patients 1 day after MRgFUS treatment [14], despite not being associated with SDR values [12]. The latter, instead, are likely to be inversely correlated with the increase in absolute energy required for a temperature rise over 37 °C [15].

Hence, considering the complex relationship between patient-related intrinsic biological characteristics (i.e., age, brain volume, skull density, cortical thickness) and MRgFUS-related parameters (i.e., SDR, ATD, T°C_max, n° of transducer elements) in determining successful thermal ablation, we sought to determine the treatment parameters that may impact the lesion volumes (necrosis and edema) with unilateral VIM thalamotomy in tremor-dominant neurological patients.

## 2. Materials & Methods

### 2.1. Patients

The participants of this study were consecutively enrolled between 19 April 2019 and 31 December 2022, as patients with medication-refractory tremor-dominant neurological disease (PD and ET) treated at IRCCS Bonino Pulejo in Messina utilizing the MRgFUS brain system. The participants in this study were recruited from the internal neurological facility. For all patients, the main clinical criterion was a clinically significant and unresponsive (to dopaminergic or anticholinergic drugs) tremor, whereas for ET patients, the integrity of the nigrostriatal dopaminergic terminals, as evidenced using a normal dopamine transporter scan (DaT-SCAN), was also considered to exclude parkinsonism. Two neurologists (G.D.L and A.Q.) with more than 20 years of expertise in movement disorders confirmed the diagnosis of either PD or ET using the IPMDS consensus criteria [16]. The exclusion criteria were (a) cognitive decline evaluated using extensive neuropsychological testing as advised by the Core Assessment Program for Surgical Interventional Therapies in Parkinson’s Disease (CAPSIT-PD) [17]; (b) unstable or severe psychiatric conditions like anxiety or depression disorder; (c) alcohol or substance abuse as defined by the Diagnostic and Statistical Manual of Mental Disorders, fifth edition (DSM-5); and (d) history of epileptic seizures.

Before starting treatment, all the patients received pre-treatment instrumental evaluations, such as head CT and brain MRI, to estimate their SDRs and rule out any organic neurological conditions that would render the treatment ineffective. The MRgFUS radiological inclusion criteria were (a) no history of intracranial hemorrhage, ischemic stroke, brain tumor, or neoplasms; (b) no intracranial aneurysms or arteriovenous malformations requiring treatment. The MRgFUS clinical exclusion criteria were (a) anticoagulant or anti-platelet therapy which the patients were not allowed to interrupt; (b) significant unstable medical conditions; (c) an SDR < 0.35 as calculated from the preliminary screening computed tomography (CT) (see below). The rationale behind this threshold was based on our own clinical experience, together with several prior studies [2,8,9,10,11]. Because the corresponding magnitude of skull density differences attenuates the ultrasonic energy that can transverse the skull, an SDR smaller than 0.4 [2,18] or 0.3 [12] is generally considered unfavorable for MRgFUS treatment [18]. In our clinical experience, we employed an SDR smaller than 0.35. This threshold may not preclude MRgFUS; nevertheless, to create a persistent thermal lesion, greater energy (i.e., sonication power, time, and number of sonications) is needed to reach a therapeutic temperature above 54 °C [6]. Such increased energy consumption raises the patient’s risk of soft tissue injury, subtherapeutic or unsuccessful treatment outcomes, and discomfort throughout the procedure. 

The internal ethics committee approved the MRgFUS VIM thalamotomy screening and surgery (CE n.38/2021), and all the patients provided written informed consent. G.D.L., A.B., C.S, and S.M. organized the internal database in which the clinical and neuroimaging data were recorded in compliance with the General Data Protection Regulation (GDPR).

### 2.2. CT Protocol

Screening CT imaging was acquired using a multidetector 64-slice CT scanner (SOMATOM Definition AS, Siemens Healthineers, Erlangen, Germany). The protocol included sequential acquisition with no gantry inclination; a tube voltage of 120 kV; a tube current of 220 mA; a slice thickness of 0.6 mm; spacing of 0 mm; reconstruction B60f sharp—Osteo. The SDR was calculated using an MRgFUS workstation (Exablate Neuro 4000, Insightec, Israel), matching the MRI and CT head images (see Appendix A Appendix A).

### 2.3. MRI Scanning Protocols

The patients underwent neuroimaging evaluation 1 day prior to treatment using a 3 T whole-body MRI scanner (Achieva, Philips Medical System; ﻿Best, Netherlands) with a 32-element phased-array sensitivity-encoding (SENSE) head coil. The MR system was equipped with gradients achieving a maximum slew rate of 200 mT/m/ms and a maximum strength of 80 mT/m. The imaging protocol included 3D T1-Weighted Magnetization-Prepared Rapid Gradient Echo (MP-RAGE), 3D Fluid-Attenuated Inversion Recovery (3D-FLAIR), 2D coronal and axial T2-weighted fast spin echo (FSE), and axial susceptibility-weighted imaging (SWI). The 3D-T1 MP-RAGE images were acquired with the following parameters: repetition time (TR): 8.2 ms; echo time (TE): 3.7 ms; section thickness: 1 mm; and reconstruction matrix: 512 × 512. The 3D-FLAIR images were acquired with a TR of 12,000 ms, a TE of 140 ms, an inversion time (TI) of 285 ms, a section thickness of 4 mm, and a reconstruction matrix = 512 × 512. The T2-weighted FSE images were acquired with a TR of 4100 ms, a TE of 100 ms, a section thickness of 2–3 mm, and a reconstruction matrix = 512 × 512. The SWI images were acquired with a TR of 31 ms, a TE of 7.2 ms, a section thickness of 3 mm, and a reconstruction matrix = 512 × 512. The MR images obtained were used for prospective preoperative planning.

### 2.4. MRgFUS Treatment

The MRgFUS thalamotomies were performed using focused ultrasound (FUS) equipment (Exablate Neuro 4000, Insightec, Haifa, Israel) integrated with a 1.5 T MRI unit (Sigma HDxt; GE Medical System, Chicago, IL, USA).

Before the procedure, the patient’s scalp was shaved and fixed with a stereotactic frame under local anesthesia. A flexible silicone membrane was positioned on the head of the patient before they entered the MRI room and laid down on the bed. The 30 cm diameter, hemispheric, 1024-element, phased-array ultrasound transducer, operating at 650 kHz (Exablate Neuro 4000, Insightec, Haifa, Israel), was connected to the attachment device of the silicone membrane. At the beginning of the procedure, an intraoperative 2 mm thickness 3D fast-recovery fast spin echo (FRFSE) T2-weighted image was acquired to estimate the stereotactic coordinates and plan the treatment target.

The treatment was planned by registering the CT and MRI images, marking as “no pass” regions any calcifications and the frontal sinuses. The number of transducer elements and the actual head surface available for the energy required to be delivered were calculated. 

Since the VIM nucleus could not be directly observed via standard MRI sequences, a combination of standardized stereotactic coordinates and neuroanatomical landmarks (based on the *Stereotactic Atlas of the Human Thalamus and Basal Ganglia* [19]) was used. The target position is calculated in intraoperative T2-weighted images 14–15 mm lateral from the midline, 10–11 mm lateral from the wall of the third ventricle, one-quarter of the total AC-PC line distance anterior of the PC, and 1.5–2 mm above the AC-PC plane (see Appendix A Appendix A). The anatomical procedure provided by Moser et al. was used as the basis for the targeting accuracy of the MRgFUS brain system used in this work [20].

To confirm accurate localization, two short (10 s) and low-power sonications (250 W) were performed to achieve a temperature rise between 40 °C and 45 °C for the alignment of the focal heating volume based on MR thermometry. The intraoperative temperature was calculated using the Exablate software (version 7.0, Insightec, Israel) based on the proton resonance frequency (PRF) shift in the water molecules (MR thermometry sequence parameters: TR/TE: 26.172 ms/12.996 ms, slice thickness: 3 mm, FOV: 28 × 28 cm^2^, matrix: 256 × 128, temporal resolution: 3.5 ms). If the alignment was correct, the high-intensity focused ultrasound beam power was then increased to achieve temperatures in the range of 50–54 °C, which resulted in a transient clinical effect. According to previous criteria [2,3,4,5,6,7,8,9,10,11,12,13,14,15], the sonication power and/or duration were then gradually increased toward the therapeutic temperature upon confirmation of the real-time clinical assessment of temporary tremor suppression or any other side effects (~55–60 °C) (see Appendix A Appendix A). At the end of the procedure, FRFSE T2-weighted images were acquired on the axial plane to visualize the resulting thalamic lesion (see Appendix A Appendix A). All the patient-related parameters were calculated during pre-treatment and the actual treatments using the Exablate workstation tools. In particular, the following parameters were considered: (a) the number of transducer elements; (b) skull area surface; (c) maximal energy delivered (Joule); (d) maximal power delivered (watt); (e) maximal sonication time (seconds); (f) T°C_max; (g) ATD. The ATD was calculated from the thermometric map of the last sonication, drawing a manual ROI at the target (see Appendix A Appendix A).

The patients were ablated in the contralateral thalamus corresponding to the dominant affected hand: a total of 16 patients were ablated in the right thalamus, while 20 patients were ablated in the left thalamus.

### 2.5. Clinical Measures

Data were collected at baseline (1 day before MRgFUS treatment) and 1 week after the treatment. All the PD patients were assessed using the Movement Disorder Society Unified Parkinson’s Disease Rating Scale motor part (MDS-UPDRS-III) [21] according to the defined OFF medication (at least 12 h after discontinuing any anti-parkinsonian medication) and ON medication conditions (90 min after a levodopa loading dose roughly equal to 150% of the patient’s usual morning dose of dopaminergic medication). Medication-resistant tremor was defined as patients who did not significantly improve between the ON and OFF medication conditions in terms of the total of their tremor subscales. The UPDRS part III and the hemi-UPDRS for the treated hemibody were used to measure changes in motor function (Items 20–26). The ET patients were assessed using the Clinical Rating Scale for Tremor (CRST) [22]. The hemi-CRST was used to evaluate changes in the treated hemibody (items 5–6, 8–9, 11–15). The tremor assessments were performed by two blinded neurologists skilled in movement disorders (A.B; G.d.L.). All the patients had undergone a pre-operative instrumental evaluation, including brain CT and MRI exams and a clinical assessment.

### 2.6. Brain Volumetric Quantification

Before the MRgFUS treatment, a volumetric analysis of patients’ white matter (WM), gray matter (GM), and cerebrospinal fluid (CSF) was performed. SPM12 (www.fil.ion.ucl.ac.uk (accessed on 1 January 2024)) was used to preprocess the structural imaging data, while the CAT12 toolbox was used to perform the volumetric analysis. The images acquired for each participant were reoriented to have the same point of origin (anterior commissure) and spatial orientation. A non-linear deformation field was estimated that best overlapped with the tissue probability maps on the individual subjects’ images. Afterward, all the native-space tissue segments were registered to the standard MNI128 using affine registration. The diffeomorphic anatomical registration through the exponentiated lie algebra (DARTEL) toolbox was applied to all the participants’ GM and WM to refine the inter-subject registration. In the last step of DARTEL, the GM/WM tissues were modulated using a non-linear deformation approach to compare the relative GM/WM volumes adjusted for individual brain size. Furthermore, the voxel values in the tissue maps were modulated using the Jacobian determinant that was calculated during spatial normalization [23]. After preprocessing, a quality check was performed using the CAT12 toolbox to assess the homogeneity of the brain tissues. Finally, each participant’s modulated and normalized brain tissue segments were smoothed using an 8 mm full-width-at-half-maximum Gaussian filter.

Additionally, the total cortical thickness estimation for all individuals was gathered using the CAT12 toolbox (http://www.neuro.uni-jena.de/cat/ (accessed on 1 January 2024)). Based on the projection-based thickness approach, the central surface and cortical thickness were estimated in one step [24]. Topology correction, spherical mapping, spherical registration, and other operations were performed [25]. The local maxima were projected onto other GM voxels using the neighbor relationship given by the WM distance after tissue segmentation.

Finally, the Fazekas scale was also used to quantify the amount of white matter T2 hyperintense lesions, usually attributed to chronic small vessel ischemia [26].

### 2.7. Lesion Volume Segmentation

After the MRgFUS treatment, semi-automated lesion size volumetry for the VIM nucleus was obtained using the open-source software ITK-SNAP version 2.2.0 (www.itksnap.org; accessed on 1 January 2024); Penn Image Computing and Science Laboratory, Philadelphia, PA, USA). The lesion volume was defined according to the co-registration of the FFE-T2 axial images with the 3D-FLAIR images in order to delineate the hypointense region related to necrosis and the hyperintensity related to edema (Figure 1).

The semi-automatic processes proceeded following previously established criteria [27]. First, after the selection of the regions of interest (ROIs), the user then converted the input picture volumes into a single synthetic image volume known as a *speed image*, using one of the presegmentation steps. Presegmentation reduces all the available image intensity voxel values to single scalar *g(x)* values, which is then referred to as the speed image. The speed image assesses the difference between the object and background probabilities at each voxel in the unsupervised classification mode. Additionally, all the available image intensity values for each voxel are used to estimate these probabilities. However, the Expectation–Maximization (EM) technique, combined with a Gaussian mixture model, yields this estimate without the need for training data [28]. After that, each image is cropped, resampled, and interpolated into the user-selected ROI space to create a smooth 3D volume. The volume calculation was performed by multiplying the number of voxels within the ROIs by the voxel size.

Two radiologists (R.M; S.C.) with more than 10 years of experience in MRgFUS practice, blinded to the subject’s identity, were responsible for semi-automatic segmentation of the lesion volumes. This measurement was performed on neuroimaging data acquired 1 week post-treatment. For each patient, the overlap of the segmented areas was calculated to measure the spatial similarity between the two raters’ segmentations in order to assess the accuracy of the manual quantification of the hypointense region (related to the necrosis volumetry) and the hyperintensity (related to the edema volumetry). To measure the coefficient of agreement between and within raters, the DICE coefficient *D* was used. The level of agreement in the volume measurements between raters was also calculated pairwise for each patient using the Intraclass Correlation Coefficient (ICC).

### 2.8. Statistical Analysis

Statistical analyses were performed using Statistical Package for Social Science software (SPSS, version 25.0, Chicago, IL, USA). The Shapiro–Wilk test was used to control the normal distribution of the data. A two-tailed significance level *p* < 0.01 was set to establish statistical significance. For exploratory purposes, statistical data surviving a one-tailed *p* < 0.05 threshold were also reported.

For the clinical data, a paired *t*-test was used to investigate significant changes after treatment. Spearman’s correlation test was used to test the relationships between all the variables and check for multi-collinearity. Finally, we sought to build a predictive model with clinical/radiological/procedural parameters and the considered lesion variables (necrosis and edema volumes) using stepwise multiple regression analysis. The rationale for including all these variables was based on the variability that has been shown in earlier research highlighting different radiological markers (such as n° of sonifications, T°C_max, and ATD) related to lesion volumetry connected to the VIM nucleus [12,13,14,15]. In this study, in addition to the well-known radiological parameters, we also included all the features that could be associated with the lesion volume induced by MRgFUS treatment, such as structural (i.e., gray/white matter volumes) and clinical factors (i.e., diagnosis, lesion side, age at treatment), in order to provide a clearer picture. This is because different statistical models were employed to evaluate the best predictors of lesion volume. Stepwise multiple regression is the step-by-step iterative construction of a regression model that involves the selection of the independent variables to be used in a final model. The removed effect is excluded from the model, and the process is repeated until no other effect in the model meets the specified level for removal. Using this approach, it was possible to evaluate the extent to which the clinical, radiological, and procedural variables might have accounted for the variation in the lesion volume and to identify the variables that contributed the most to the explanation of variance. The probability for entry into stepwise regression was set at 0.05.

## 3. Results

### 3.1. Brain Structural and Radiological Data

Thirty-six neurological patients with hemi-dominant tremor satisfied the clinical criteria and were assigned to MRgFUS treatment (19 PD and 17 ET) (PD: 68.4% male; mean age: 69.1 ± 7.6 years; mean disease duration: 8.8 ± 5.1 years; ET: 58.8% male; mean age: 67.7 ± 10.1 years; mean disease duration: 18.7 ± 15.5 years). A pre-treatment MRI exam was employed for brain volumetric quantification (see Table 1). Before treatment, no significant differences were detected between the ET and PD patients in brain volumetry (see Appendix A).

Table 2 shows the overall MRI-based procedural parameters employed during VIM thalamotomy in the neurological patients. The enrolled patients had a small number of frequently reported side effects over the course of the treatment (see Appendix A). One week after the treatment, an additional MRI exam was performed in order to calculate the lesion size in the VIM nucleus (Table 3). The data analysis of the necrosis and edema measurements showed optimal agreement in terms of both inter-rater and intra-rater agreement (Table 3). Moreover, the mean and standard deviation of the DICE coefficient were optimal at 0.74 ± 0.13 and 0.7 ± 0.11, respectively, for the edema and necrosis volumetry quantifications (Figure 2).

### 3.2. Clinical Data

Following FUS ablation, all patients showed significant motor improvement (all t-values > 10; all *p*-values < 0.001) and a reduction in dominant hand tremor at 1 week after the treatment with respect to the baseline (all t-valued > 11; all *p*-values < 0.001) (See Appendix A, Figure 3). The clinical recovery was similar for the PD and ET patients, as well as for the patients who underwent left or right VIM ablation (see Appendix A).

### 3.3. Regression Analyses

Among the MRgFUS parameters, significant associations were detected (for more details, See Appendix A). The demographic and clinical scores recorded before treatment showed significant relationships with the brain volumetric data and MRgFUS parameters. In particular, the variable age was negatively correlated with the Normalized GM Volume (Rho = −0.65, *p* < 0.001) and positively correlated with the Normalized WM Volume (Rho = 0.41, *p* = 0.01). Moreover, as concerns the brain lesion measures (necrosis and edema volumes), several correlations were found. The necrosis volume was negatively correlated with n° of transducer elements (Rho = −0.48, *p* = 0.003) and Normalized GM Volume (Rho = −0.33, *p* = 0.04) and positively correlated with Normalized WM Volume (Rho = 0.35, *p* = 0.04) and T°C_max (Rho = −0.37, *p* = 0.02). Edema volume was positively correlated with the SDR (Rho = 0.36, *p* = 0.02) and T°C_max (Rho = 0.51, *p* = 0.002) and negatively with n° of transducer elements (Rho = −0.36 *p* = 0.03) and Majoue (Rho = −0.33 *p* = 0.04). Finally, we also detected significant relationships between clinical improvement after treatment (delta scores for the UPDRS and CRST scales) and the radiological parameters (for additional information, see Appendix A).

A multiple linear regression model was employed to identify the factors that may predict lesion volume (necrosis and edema). No collinearity was observed among the selected variables (i.e., tolerance ≥ 0.99, VIF ≤ 1.01). For the necrosis volume, regression analysis provided three specific model of predictors. In model n°3, n° of transducer elements, T°C_max, and Normalized WM Volume were significantly reported as the best predictors (F3,32 = 8.39, *p* = 0.0001, R2 = 0.44). Similarly, for the edema volumes, model n°2 included T°C_max and n° transducer elements (F_2,33_ = 9.60, *p* = 0.001, R^2^ = 0.37) as the best predictors. Table 4 showed the coefficients of the model and the results of the *t*-test, used to study the significance of the regression coefficients (βi) for the necrosis and edema volumes. 

## 4. Discussion

Our study shows that the mean maximal temperature reached (T°C_max) and number of transducer elements are the best predictors of treatment-induced necrosis and edema volumes, regardless of the various MRgFUS-related parameters used during the thermal ablation of the VIM nucleus in tremor-dominant neurological patients. Moreover, necrosis volume was also significantly associated with WM volume. In comparison to the previous research, our study offers two significant methodological innovations: (a) the use of objective and semiautomated 3D volumetric measurements of thalamotomy lesion size and (b) the evaluation of multiple factors, including clinical, neuroimaging, procedural, and demographic parameters, to assess their combined effects on lesion volume. 

The first main finding of our study is that the variation in T°C_max and the number of transducer elements predicts the lesion volumes of edema and necrosis. Reaching therapeutic temperatures at the target is a critical component of MRgFUS treatment. The degree of T°C_max and the max sonification time are the two main factors that typically determine the extent of thermal lesioning [29] and are perfectly correlated [30] (see Appendix A Appendix A). In our dataset, the mean T°C_max was > 57° (ranging from 51° to 63°), in perfect agreement with previous studies [31]. In particular, Gagliardo et al. [7] found that there is a strong correlation between the intraoperatively determined lesion diameter of zones I and II and T°C_max and the mean average temperatures per sonication. However, a higher target temperature results in larger lesions [30], but larger lesions may increase the risk of adverse effects [32,33]. Therefore, to reach a good therapeutic impact and prevent the transient collateral side consequences brought on by edema, it is crucial to confine necrosis, which must be precisely localized at the target (VIM). To reach this goal, several treatment parameters needed to be controlled. Skull volume and the SDR as well as the number of transducer elements used have been demonstrated to correlate with the highest temperatures reached using MR thermometry, though the precise relationship between the acoustic energy and the reached temperatures is still unclear [27,31]. Our simple regression analysis revealed that the SDR and T°C_max were positively correlated with edema volume. According to Chang et al. [9], a high SDR is necessary to obtain higher temperatures and more lesioning. However, our multiple regression analysis revealed that the n° of transducers elements was the only factor determining the final lesion volumes. The number of transducer elements plays a role in how effectively and precisely heat is delivered. An effective treatment usually needs at least 700 elements out of the maximal 1024 and is usually carried out using 850 to 950 elements [34]. In our dataset, the mean number of elements was 906.6, showing a negative relationship with both lesion volumes. To the best of our knowledge, this is the first study highlighting the role of this procedural parameter in determining the final lesion size. Our data would seem to suggest that the relationship between the number of transducer elements and other MRgFUS-related parameters in determining the final lesion size may be not linear, probably depending on multiple factors. Indeed, it involves balancing the need for precise targeting (i.e., the spatial resolution of the focused ultrasound, the distribution of the ultrasound transducers), efficient energy delivery (i.e., n° of sonications, skull area, maximal energy and power delivered, maximal sonication time delivered), individual brain characteristics (skull density, brain anatomical abnormalities, calcification), and safety considerations. 

Another piece of evidence highlighted by the multiple regression analysis concerns the association between the total volume of white matter and the volume of necrosis. Ultrasounds are mechanical waves that can be reflected, refracted, or attenuated before reaching the target, and the propagation speed strongly varies according to the skull bone volume, as well as the density of every single intervening tissue (GM, WM, and CSF). Indeed, it has been demonstrated that these factors can deeply influence the amount of energy carried by the acoustic wave that is transmitted rather than reflected or absorbed [5,6,7,14,34]. For this reason, our study is perfectly in agreement with the previous literature, adding a new piece of information about the influence of the total white matter volume as a predictive factor for the necrosis volume.

### Limitations

Some limitations of this exploratory study should be highlighted. The lesion volume changes were assessed 1 week after the MRgFUS treatment. As a result, additional neurobiological consequences of potential restoration may have not yet been analyzed (mostly in the case of edema volumetry). Further studies with longer follow-up periods are needed. Next, we investigated tremor-dominant neurological patients, merging PD with ET patients, who are characterized by different neurodegenerative processes that would impact the effectiveness of MRgFUS treatment, as well as the neurobiological response to ultrasound. Finally, the apparent discrepancy between our results with respect to the previous literature highlighting the role of other treatment parameters in determining the final lesion volume (i.e., ATD [14]) can be justified by one specific methodological difference. Indeed, only a few studies have distinguished the lesion volumes between necrosis and edema areas. Moreover, the significant correlations reported in the previous literature mainly refer to 2D lesion measurement (area, shape, dimension) or 2D total lesion size. Otherwise, a 3D semi-automated volumetric assessment of the two distinct lesion areas and an evaluation of the relationship with neuroimaging, clinical, and procedural parameters have never been performed. However, a tailored and shared 3D volumetric approach to lesion volume assessment in thalamic ablation following MRgFUS treatment is still pending validation by the neuroimaging community.

## 5. Conclusions

The procedural efficacy of MRgFUS thalamotomy in treating tremor-dominant PD and ET patients is strictly dependent on the production of an appropriately sized lesion, which is important for achieving durable clinical effects [35,36]. Therefore, it is essential to understand the relationship between the factors (patient-related intrinsic biological characteristics and MRgFUS parameters) that may impact the lesion volume following MRgFUS treatment. Despite the clinical heterogeneity in the enrolled neurological population, the present study highlights that T°C_max and the number of transducer elements are the best radiological parameters to consider during an MRgFUS procedure. In particular, smaller volumes of the necrosis and edema, smaller maximal temperatures, and a higher number of transducer elements are important to note. Moreover, the total white matter volume at baseline would seem to impact the necrosis volume. Future longitudinal and multicentric studies are required to ascertain the multidimensional link between these parameters and lesion volume in neurological tremor-dominant patients after MRgFUS VIM therapy given the significant discrepancy in the literature’s findings. Simple or multiple regression statistical analysis is not enough to represent the complexity of the multidimensional relationship between patient-related intrinsic biological characteristics and MRgFUS-related parameters (i.e., SDR, ATD, T°C_max, n° of transducer elements) in determining successful thermal ablation. The employment of more sophisticated statistical approaches (structural equation modeling, principal component analysis, machine learning, or radiomics [37,38]) is strongly suggested for future studies.

## Figures and Tables

**Figure 1 bioengineering-11-00373-f001:**
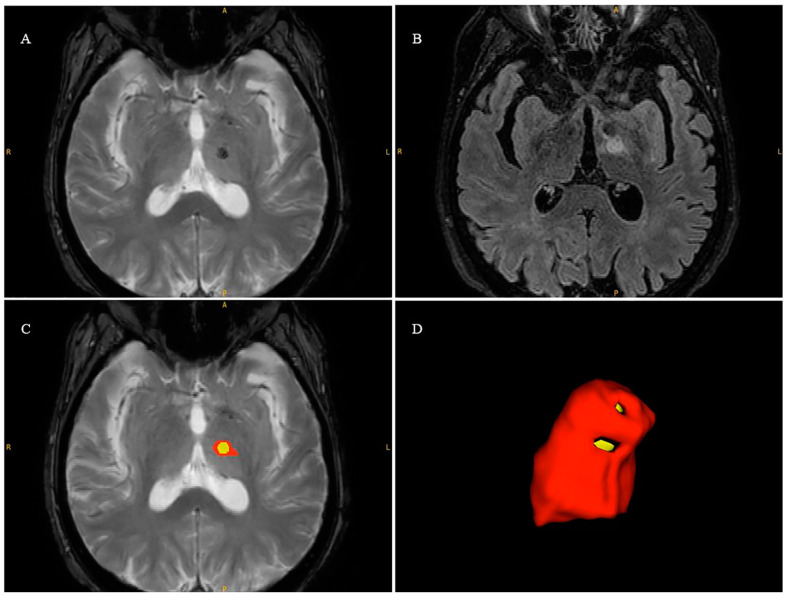
Three-dimensional reconstruction of VIM lesion induced by MRgFUS treatment. (**A**) FFE-T2 axial image. (**B**) A 3D-FLAIR axial image. (**C**) The necrosis (yellow area) and edema (red area) semi-automated segmentation superimposed onto the FFE-T2 image co-registered with the 3D-FLAIR image. (**D**) The 3D volumetric reconstructions of necrosis and edema made using ITK-SNAP software (V 4.2).

**Figure 2 bioengineering-11-00373-f002:**
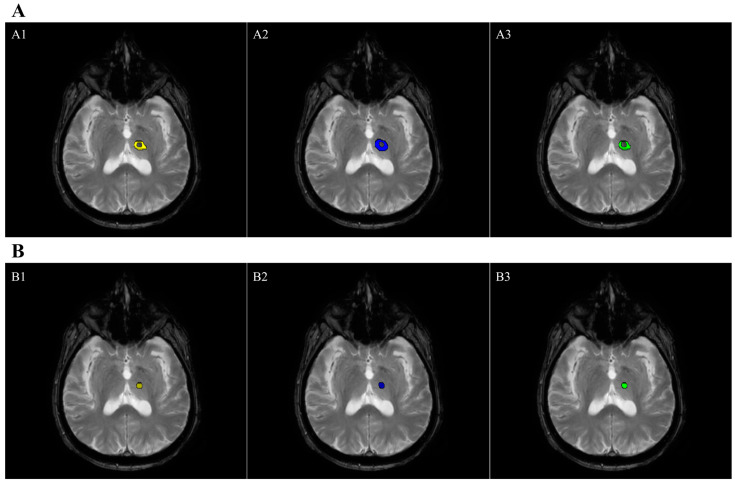
Lesion segmentation of a neurological patient superimposed onto an FFE-T2 image co-registered with a 3D-FLAIR image. Axial plane view (the right hemisphere is depicted on the left side). (**A**) Segmentation of edema: (**A1**) = rater 1 edema segmentation (yellow color); (**A2**) = rater 2 edema segmentation (blue color); (**A3**) = overlapping of the two edema segmentations (green color). (**B**) Segmentation of necrosis: (**B1**) = rater 1 necrosis segmentation (yellow color); (**B2**) = rater 2 necrosis segmentation (blue color); (**B3**) = overlapping of the two necrosis segmentations (green color).

**Figure 3 bioengineering-11-00373-f003:**
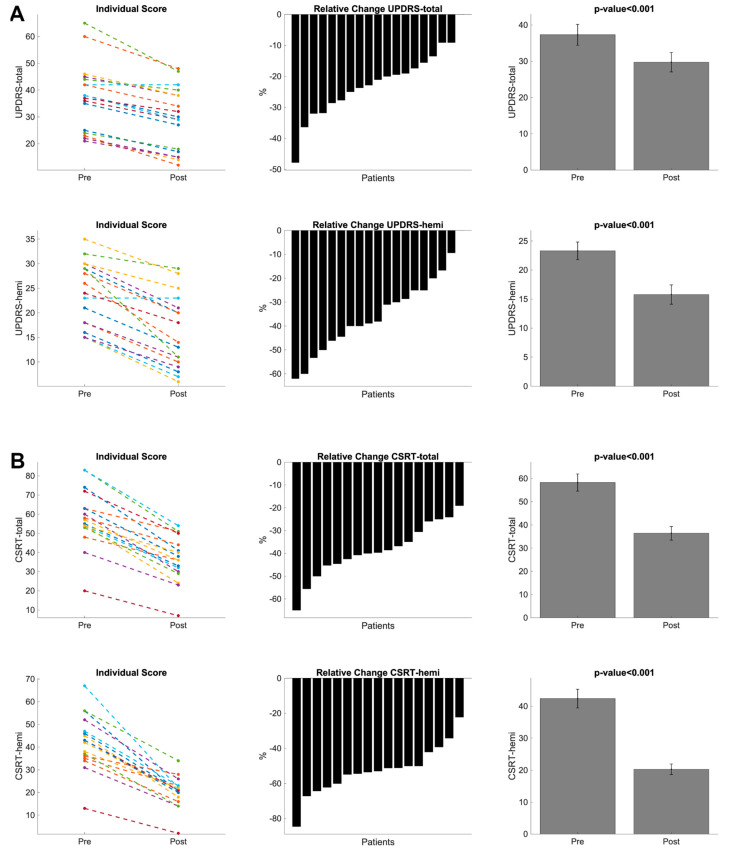
Summary of clinical scores before and after MRgFUS treatment. Patients with tremor-dominant Parkinson’s disease (**A**) and essential tremor (**B**). Movement Disorder Society Unified Parkinson’s Disease Rating Scale motor part (MDS-UPDRS-III). Clinical Rating Scale for Tremor (CRST).

**Table 1 bioengineering-11-00373-t001:** Mean brain structural characteristics of the neurological population before MRgFUS treatment.

Tremor-Dominant Patients (N°36)	Mean ± SD
Normalized GM Volume (cm^3^)	326.3 ± 35.3
Normalized WM Volume (cm^3^)	292.8 ± 41.3
Normalized CSF Volume (cm^3^)	377.5 ± 49.6
Normalized Brain Volume (GM + WM) (cm^3^)	619.1 ± 48.4
Total Cortical Thickness (mm)	2.61 ± 0.2
Fazekas Scale	3 [0–7]

GM: gray matter; WM: white matter; CSF: cerebrospinal fluid. Data are reported as mean or median values.

**Table 2 bioengineering-11-00373-t002:** MRgFUS procedural parameters used during VIM thalamotomy.

Tremor-Dominant Patients (N°36)	Mean ± SD Values
SDR	0.49 ± 0.12
Skull Area (cm)	338.6 ± 31.3
N° of Transducer Elements	906.6 ± 66.2
N° of Sonications	6.9 ± 2.1
Maximal Watt	896.3 ± 139.5
Maximal Joule	17673.6 ± 7413.7
Maximal Temperature (T°C_max)	57.4 ± 3.4
Maximal Sonication Time (s)	22.9 ± 7.5
ATD	56 ± 2.9

SDR: skull density ratio; ATD: accumulated thermal dose. Data are reported as mean values.

**Table 3 bioengineering-11-00373-t003:** Brain lesion volume quantification.

	I° Rater	II° Rater	ICC	Mean Dice
Edema Volume (cm^3^)	1607.24 ± 705.5	2042.76 ± 895.92	0.81	0.74 ± 0.13
Necrosis Volume (cm^3^)	179.31 ± 135.72	153.35 ± 118.56	0.88	0.70 ± 0.11

**Table 4 bioengineering-11-00373-t004:** Multiple regression analysis.

Predictors for Necrosis Volume						
N° Model	R	R^2^	Variable	Standardized β Coefficients	*t*-Value	*p*-Value
Model 1	0.484	0.234	N° transducer elements	−0.484	−3.22	0.003
Model 2	0.595	0.354	N° transducer elements	−0.466	−3.32	0.002
			T°C_max	0.346	2.47	0.02
Model 3	0.664	0.44	N° transducer elements	−0.441	−3.32	0.002
			T°C_max	0.345	2.61	0.01
			Normalized WM Volume	0.295	2.23	0.03
**Predictors for Edema Volume**						
Model 1	0.508	0.258	T°C_max	0.508	3.44	0.002
Model 2	0.606	0.368	T°C_max	0.491	3.54	0.001
			N° transducer elements	−0.331	−2.39	0.023

## Data Availability

The datasets associated with the present study are available upon reasonable request by interested researchers.

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
