# Peer review of "The Role of Treatment-Related Parameters and Brain Morphology in the Lesion Volume of Magnetic-Resonance-Guided Focused Ultrasound Thalamotomy in Patients with Tremor-Dominant Neurological Conditions"

_bioengineering, 2024, doi:10.3390/bioengineering11040373_

Round 1

Reviewer 1 Report

Comments and Suggestions for Authors

The paper requires major revisions.

1. How did you ensure the selection criteria for patients with medication-refractory tremor were met consistently across both Parkinson's disease (PD) and Essential Tremor (ET) groups?

2. Could you elaborate on the rationale behind choosing a specific cutoff value for the Skull Density Ratio (SDR) in the radiological inclusion criteria?

3. In the pre-treatment MRI evaluations, were there any notable differences in brain structural characteristics between PD and ET patients?

4. Can you provide more insight into how the stereotactic coordinates were determined for targeting the VIM nucleus, especially considering the variation in brain anatomy among patients?

5. Was there any consideration given to the potential impact of variations in skull thickness and density on the accuracy of targeting and lesion formation?

6. Could you explain further the rationale behind selecting specific procedural parameters such as sonication power, duration, and the number of transducer elements?

7. How were the specific thresholds for temperature and sonication duration determined to achieve precise necrosis during the procedure?

8. Were there any observed differences in treatment outcomes between patients who underwent right thalamus ablation versus left thalamus ablation?

9. In the clinical assessments, were there any significant differences in motor improvement between PD and ET patients, considering the different assessment scales used?

10. Could you discuss any unexpected findings or challenges encountered during the post-treatment MRI evaluations for lesion volume quantification?

11. What measures were taken to ensure consistency and reliability in the semi-automated lesion volume segmentation process performed by the radiologists?

12. Were there any observed correlations between demographic or clinical factors and the MRgFUS procedural parameters?

13. Can you explain the rationale behind choosing specific variables for inclusion in the multiple linear regression models for predicting lesion volumes?

14. Were there any limitations or assumptions made in the regression analyses that might affect the generalizability of the predictive models?

15. How do the identified predictors of lesion volume (necrosis and edema) align with previous literature or clinical expectations in the field?

16. Were there any unexpected or counterintuitive findings in the regression analyses that merit further investigation?

17. What implications do the identified predictors of lesion volume have for clinical practice, particularly in terms of treatment planning and patient outcomes?

18. Could you discuss any potential future research directions or areas for further investigation based on the findings of this study? Add suitable references to the site with [PMID: PMID: 34544468, PMID: 37993801]

19. In conclusion, how do the results of this study contribute to our understanding of MRgFUS thalamotomy and its application in patients with medication-refractory tremor?

Author Response

The paper requires major revisions.

  1. How did you ensure the selection criteria for patients with medication-refractory tremor were met consistently across both Parkinson's disease (PD) and Essential Tremor (ET) groups?

REPLY: Following the reviewer’s suggestion this section has been modified as follows: “Participants in this study were consecutively enrolled between April 19, 2019, and December 31, 2022, as patients with medication-refractory tremor dominant neurological disease (PD and ET) treated at IRCCS Bonino-Pulejo in Messina utilizing the MRgFUS brain system. The participants in this study were recruited from the internal neurological facility. For all patients, the main clinical criterion was a clinically significant and unresponsive (to dopaminergic or anticholinergic drugs) tremor, whereas for ET patients the integrity of the nigrostriatal dopaminergic terminals, as evidenced by a normal dopamine transporter scan (DaT-SCAN), was also considered to exclude parkinsonisms. Two neurologists (G.D.L & A.Q.) with more than 20 years of expertise in movement disorders confirmed the diagnosis of either PD or ET using the IPMDS consensus criteria [16].”

  1. Could you elaborate on the rationale behind choosing a specific cutoff value for the Skull Density Ratio (SDR) in the radiological inclusion criteria?

REPLY: We express our gratitude to the reviewer for drawing attention to this specific query. We sincerely regret the mistake that was made earlier. Naturally, one of the exclusion criteria was patients with SDR< 0.35. For this reason, we revised this part as follows to your suggestion: Before starting treatment, all patients received pre-treatment instrumental evaluations, such as head CT and brain MRI, to estimate the SDR and rule out any organic neurological conditions that would render the treatment ineffective. MRgFUS radiological inclusion criteria were: a) no history of intracranial hemorrhage, ischemic stroke, brain tumour, or neoplasms; b) no intracranial aneurysms or arteriovenous malformations requiring treatment. MRgFUS clinical exclusion criteria were: a) anticoagulant or anti-platelet therapy which the patients were not allowed to interrupt; b) significant unstable medical conditions; c) skull density ratio (SDR) < 0.35 as calculated from the preliminary screening computed tomography (CT)(see below). The rationale behind this threshold is based on our own clinical experience together with several prior studies [2, 8–11]. Because the corresponding magnitude of skull density differences attenuates the ultrasonic energy that would transverse the skull, an SDR smaller than 0.4 [2, 18] or 0.3 [12] is generally considered unfavorable for MRgFUS treatment [18]. In our clinical experience we employed an SDR smaller than 0.35. This threshold may not preclude MRgFUS; nevertheless, to create a persistent thermal lesion, greater energy (i.e., sonication power, time, and number of sonications) is needed to reach a therapeutic temperature above 54°C [6]. Such increased energy consumption raises the patient's risk of soft-tissue injury, subtherapeutic or unsuccessful treatment outcomes, and discomfort throughout the procedure.”

  1. In the pre-treatment MRI evaluations, were there any notable differences in brain structural characteristics between PD and ET patients?

REPLY: We would like to thank this reviewer for highlighting this important point. We now included a new statement in the results section as well as a new table (S1) in the supplementary materials in order to show the perfect similarity between ET’ and PD’ brains.

  1. Can you provide more insight into how the stereotactic coordinates were determined for targeting the VIM nucleus, especially considering the variation in brain anatomy among patients?

REPLY: Following the reviewer’s suggestion we formulated this section as follows: “Since the VIM nucleus could not be directly observed via standard MRI sequences, a combination of standardized stereotactic coordinates and neuroanatomical landmarks (based on the Stereotactic Atlas of the Human Thalamus and Basal Ganglia [18]) were used. Target position is calculated on intraoperative T2-weighted images: 14-15 mm lateral from the midline, 10-11 mm lateral from the wall of the third ventricle, one quarter of the total AC-PC line distance anterior of the PC and 1,5-2 mm above the AC-PC plane (see Suppl. Materials Fig.S2). Overall, the anatomical processes offered by Moser et al. have served as a basis for the targeting accuracy of the MRgFUS brain system used in this work. [19]”

  1. Was there any consideration given to the potential impact of variations in skull thickness and density on the accuracy of targeting and lesion formation?

REPLY: the degree of temperature increase during thermal lesioning is critical to the efficacy of the treatment. The degree of Tmax and the duration of Tmax are the two criteria that typically determine the extent of thermal lesioning. For instance, tissue necrosis at 57°C is known to occur in 1 second, but tissue necrosis at 54°C requires 3 seconds. Like other thermal lesioning techniques, MRgFUS could potentially change the Tmax duration by prolonging the sonication period. Even with greater sonication energies, the degree of Tmax revealed limitations.  According to Chang et al., (2015) study, energy transfer across the skull is more linked with patient-related characteristics—namely, SDR and skull volume—than with treatment-related parameters. For this reason, the relationship between individual skull thickness and volume, energy delivered, and the efficacy of thermal lesioning was determined according to international guidelines.

With this in mind, we formulated the 2.4 section as follows: According to previous criteria [2,6,9] sonication power and/or duration were then gradually increased toward therapeutic temperature upon confirmation of the real-time clinical assessment of temporary tremor suppression or any other side effects (~55°-60°C)”

  1. Could you explain further the rationale behind selecting specific procedural parameters such as sonication power, duration, and the number of transducer elements?

REPLY: We have already described the established procedure for explaining how treatment parameters contributed to the accumulation of ablative thermal dose per sonication.

  1. How were the specific thresholds for temperature and sonication duration determined to achieve precise necrosis during the procedure?

REPLY: To induce necrosis, one must reach temperature spikes above at least 56-58° maintained for about 2-3 sec. The total sonication time varies about the joules intended to be delivered depending on watts and time e.g. 12,000 joules = 800 watts x 15 sec or 900 watts x 13 sec. However, these are standard procedural parameters.

  1. Were there any observed differences in treatment outcomes between patients who underwent right thalamus ablation versus left thalamus ablation?

REPLY: We would like to thank this reviewer for highlighting this important point. No relevant clinical differences were detected during the treatment as demonstrated by the prevalence of side effects. As concerns clinical outcome we included a new Table S3 describing all clinical evidence considering the entire group divided for right and left thalamus ablation.

9) In the clinical assessments, were there any significant differences in motor improvement between PD and ET patients, considering the different assessment scales used?

REPLY: Either PD or ET patients had the same level of clinical improvement, as seen in Table S3.

  1. Could you discuss any unexpected findings or challenges encountered during the post-treatment MRI evaluations for lesion volume quantification?

REPLY: No relevant expected finding was detected during volume quantification. However, a new statement has been included in the limitation sections. Moreover, the significant correlations reported in the previous literature mainly refer to 2D lesion measurement (area, shape, dimension) or 2D total lesion size. Otherwise, a 3D semi-automated volumetric assessment of the two distinct lesion areas and an evaluation of the relationship with neuroimaging, clinical and procedural parameters has never been performed. However, a tailored and shared 3D volumetric approach for lesion volume assessment of thalamic ablation following MRgFUS treatment is still pending validation by the neuroimaging community.”

  1. What measures were taken to ensure consistency and reliability in the semi-automated lesion volume segmentation process performed by the radiologists?

REPLY: We now better explain this analysis in the section 2.7

Two radiologists (R.M; S.C.) with more than 10 years of experience in MRgFUS practice, blind to the subject’s identity, were responsible for the semi-automatic segmentation of lesion volumes. This measurement was performed on neuroimaging data acquired 1-week post‐treatment. For each patient, the overlap of the segmented areas was performed to measure the spatial similarity between the two raters' segmentations in order to assess the accuracy in manual quantification of the hypointense region (related to necrosis volumetry) and the hyperintensity (related to edema volumetry). To measure the coefficient of agreement between and within raters the DICE coefficient D was used. The level of agreement for volume measurements between raters was also calculated pairwise for each patient by Intraclass Correlation Coefficient (ICC).”

  1. Were there any observed correlations between demographic or clinical factors and the MRgFUS procedural parameters?

REPLY: Following the reviewer’s suggestion we now included new tables in the supplementary materials (Table S5) to better provide a general picture of our findings. The results section has been modified accordingly. 

  1. Can you explain the rationale behind choosing specific variables for inclusion in the multiple linear regression models for predicting lesion volumes?

REPLY: Following the reviewer’s suggestion we now re-formulated this part as: For clinical data, a paired t-test was used to investigate significant changes after treatment.  The Spearman correlation test was used to test the relationships between all variables and check for multi-colinearity. Finally, we sought to build a predictive model with clinical/radiological/structural/procedural parameters and the considered lesion variables (necrosis and edema volumes) by using stepwise multiple regression analysis. The rationale for including all these variables is based on the variability that has been shown in earlier research highlighting different radiological markers (such as n° sonifications, T°C max, and ATD) related to lesion volumetry connected to VIM [12–15]. In this study, in addition to well-known radiological parameters, we also included all features that could be associated with lesion volume induced by MRgFUS treatment, such as structural (i.e., gray/white matter volumes) and clinical factors (i.e., diagnosis, lesion side, age at treatment) in order to provide a clearer picture. This is because different statistical models were employed to evaluate the best predictors of lesion volume. Stepwise multiple regression is the step-by-step iterative construction of a regression model that involves the selection of independent variables to be used in a final model. The removed effect is excluded from the model, and the process is repeated until no other effect in the model meets the specified level for removal.”

  1. Were there any limitations or assumptions made in the regression analyses that might affect the generalizability of the predictive models?

REPLY: The limitations of our statistical approach have already been discussed in the limitation and conclusion sections.

  1. How do the identified predictors of lesion volume (necrosis and edema) align with previous literature or clinical expectations in the field?

REPLY: The main innovations of our study concerning previous literature are summarized in these 3 points:

  1. First of all, we provide the best predictors of MRgFUS VIM-related lesion volume, dividing edema, and necrosis volumetry using a 3D semi-automated volumetric approach. In most of the previous studies, lesion volumes were considered together (total lesion volume merging edema and necrosis) and in a 2D space (concerning a more informative 3D space).
  2. The reported relationship between T°C_max/n° of transducer elements and edema/necrosis volumes was perfectly in agreement with previous literature (23-25] despite the employment of different statistical and volumetry quantification approaches. The role of these two radiological parameters is well-known in the neuroimaging community. Instead, the new finding is that this relationship persisted also considering a vast number of clinical and other radiological parameters.
  3. Last but not least, another advancement concerning previous literature concerns the involvement of total white matter volume that could affect the detected lesion volume. This additional finding is clinically meaningful since the effectiveness of sonification strongly depends on the integrity of white matter compartments.

All these considerations have already been included in the discussion section.

  1. Were there any unexpected or counterintuitive findings in the regression analyses that merit further investigation?

REPLY: As stated in the Conclusions section, our regression analysis provides a clearer picture of the best predictors of MRgFUS VIM-related lesion volume, since we pooled together many potential variables that might influence this treatment. We confirm part of the previous literature adding new insights on the role, for instance, of the Total White Matter Volume. Thus, we believe that our analysis can be considered a reliable scientific advancement, despite the clinical limitations (merging PD and ET patients together). However, the main innovation for further investigation will be the employment of more sophisticated statistical approaches (structural equation modeling, principal component analysis and machine learning) that will shed new light on the intimate relationship between this complex multidimensional phenomenon.  

  1. What implications do the identified predictors of lesion volume have for clinical practice, particularly in terms of treatment planning and patient outcomes?

REPLY: As stated in the Conclusions section, after longitudinal validation of our evidence and the implementation of different statistical approaches (i.e. Structural Equation Modeling) we will be able to provide new reliable radiological markers that could be employed in the clinical practice to better predict lesion size, thus improving the procedures’ efficacy of MRgFUS thalamotomy.

  1. Could you discuss any potential future research directions or areas for further investigation based on the findings of this study? Add suitable references to the site with [PMID: PMID: 34544468, PMID: 37993801]

REPLY: Done

  1. In conclusion, how do the results of this study contribute to our understanding of MRgFUS thalamotomy and its application in patients with medication-refractory tremor?

REPLY: As already stated in the Introduction, the rationale behind this kind of research is to improve our understanding of the complex relationship between patient-related intrinsic biological characteristics (i.e., age, brain volumes, skull density, cortical thickness) and MRgFUS-related parameters (i.e., SDR, ATD, T°C_Max; n° of transducer elements) in determining the successful thermal ablation.

Reviewer 2 Report

Comments and Suggestions for Authors

The title of the paper proposed by the author is interesting but there is some flaws listed out below:

1. This work is proposed for prediction of Parkinson’s disease (PD) and Essential Tremor (ET) patient by using medical image processing, include the significance or motivation of the research in Introduction section.

2. Emphasize the recent literature related to this research work and include the block diagram for the proposed work.

3. The data collection is one the important step, have to explain about the database where it is taken and full specification should present.

4. 3D reconstruction of VIM lesion induced by MRgFUS treatment is obtained in the figure but there is no proper algorithm or mathematical expression related to this work.

5. The brain tumor is the main work of this research but there is a lot of research gap and table 1 & 2 results is not clearly explained. 

6. Tremor Dominant Patients (N°36) result table obtain the mean + SD, there is no proper explain why it is alone essential.

7. The segmentation of this work, Lesion volume quantification. Segmentation overlapping of a neurological patient super- 279 imposed on the FFE-T2 images. A) Segmentation of edema between rater I (red blob) and rater II 280 (blue blob). B) Overlapping of the two edema segmentations. C) Segmentation of necrosis between 281 rater I (red blob) and rater II (blue blob). D) Overlapping of the two necrosis segmentations, is presented. But the content related to this is not clear and there is no novelty in the research.

8. Predictors for Necrosis volume three model is taken & Edema volume two model is taken. Why it is two different number of models taken and validated this process? How we claim for validation of the proposed work?

9. Based on the changes in the manuscript, revise the abstract and conclusion. Ensure the language the manuscript once or twice with the respective language speakers. 

Comments on the Quality of English Language

Ensure the language of the entire paper and avoid typos errors

Author Response

The title of the paper proposed by the author is interesting but there is some flaws listed out below:

1) This work is proposed for the prediction of Parkinson’s disease (PD) and Essential Tremor (ET) patients by using medical image processing, include the significance or motivation of the research in Introduction section. Emphasize the recent literature related to this research work and include the block diagram for the proposed work.

REPLY. The neuroimaging evaluation of MRgFUS treatment of VIM ablation for tremor disorders in neurological patients is a very restricted field of study. In the introduction (as well as in the discussion) the most relevant and recent works in this field of study have already been included.

2) The data collection is one the important step, have to explain about the database where it is taken and full specification should present.

REPLY: Following the reviewer’s suggestion, we now included a new statement in the Methods section ”Participants in this study were consecutively enrolled between April 19, 2019, and December 31, 2022, as patients with medication-refractory tremor dominant neurological disease (PD and ET) treated at IRCCS Bonino-Pulejo in Messina utilizing the MRgFUS brain system. The participants in this study were recruited from the internal neurological facility.

“The internal Ethics Committee approved the MRgFUS VIM thalamotomy screening and surgery (CE n.38/2021), and all patients provided written informed consent. G.D.L., A.B, C.S and S.M. organized the internal database where clinical and neuroimaging data were recorded in compliance with the General Data Protection Regulation (GDPR).

3) 3D reconstruction of VIM lesion induced by MRgFUS treatment is obtained in the figure but there is no proper algorithm or mathematical expression related to this work.

REPLY: In agreement with the reviewer’s suggestion, we now re-formulated the description of our semi-automated approach: The semi-automatic processes proceed following previously established criteria [23]. First, after the selection of the region-of-interests (ROI), the user then converted the input picture volumes into a single synthetic image volume known as the speed image, using one of the presegmentation steps. Presegmentation reduces all the available image intensity values at a voxel to a single scalar g(x) value, which is then referred to as the speed image. The speed image assesses the difference between object and background probability at each voxel in the unsupervised classification mode. Additionally, all available image intensity values for each voxel are used to estimate these probabilities. However, the Expectation-Maximization (EM) technique combined with a Gaussian mixture model yields this estimate without the need for training data [24]. After that, each image is cropped, resampled, and interpolated into the user-selected ROI space to create a smooth 3D volume. Volume calculation was performed by multiplying the number of voxels within ROIs X voxels size.”

  1. The brain tumor is the main work of this research but there is a lot of research gap and table 1 & 2 results is not clearly explained. 

REPLY: We regret to inform you that the clinical goal of this work is to apply a well-validated neurosurgical technique for thermal ablating VIM in individuals with tremor-dominant neurological disorders. This branch of study has nothing to do with brain tumors. According to the clinical criteria, having a brain tumor is an exclusion factor.

  1. Tremor Dominant Patients (N°36) result table obtain the mean + SD, there is no proper explain why it is alone essential.

REPLY: In tables 1-3 we reported the main findings related to neuroimaging, and procedural MRI data considering the entire neurological group. We now included several supplementary tables (see supplementary file) to better describe the clinical and neuroimaging data according to different clinical phenotypes (ET versus PD).

  1. The segmentation of this work, Lesion volume quantification. Segmentation overlapping of a neurological patient superimposed on the FFE-T2 images. A) Segmentation of edema between rater I (red blob) and rater II 280 (blue blob). B) Overlapping of the two edema segmentations. C) Segmentation of necrosis between rater I (red blob) and rater II (blue blob). D) Overlapping of the two necrosis segmentations, is presented. But the content related to this is not clear and there is no novelty in the research.

REPLY: we perfectly agree with this reviewer. For this reason, a new figure and legends have been included.

  1. Predictors for Necrosis volume three model is taken & Edema volume two model is taken. Why it is two different number of models taken and validated this process? How we claim for validation of the proposed work?

REPLY: As usual, we chose the best model where the R2 was maximal considering separately the regression model for predicting edema lesion volume and necrosis lesion volume. The statistical analysis section has been re-formulated accordingly.

  1. Based on the changes in the manuscript, revise the abstract and conclusion. Ensure the language the manuscript once or twice with the respective language speakers. 

REPLY: A native English speaker has revised the entire manuscript. However, we regret to notify you that, although we have improved the methodology section and clinical findings following all reviewers' suggestions, the abstract and conclusion do not need to be modified. Indeed, we believe that there is no justification for changing the abstract or conclusion.

Round 2

Reviewer 1 Report

Comments and Suggestions for Authors

The article revised by the author according to the comments of the reviewers basically meets the requirements.